# A Review of Chemicals to Produce Activated Carbon from Agricultural Waste Biomass

**Kalu Samuel Ukanwa [1], Kumar Patchigolla [1,\*], Ruben Sakrabani [2], Edward Anthony [1]**  **and Sachin Mandavgane [3]**

[1] Centre for Thermal Energy and Materials, School of Water, Energy and Environment, Cranfield University, Cranfield MK43 0AL, UK; Kalu.Ukanwa@cranfield.ac.uk (K.S.U.); b.j.anthony@cranfield.ac.uk (E.A.)
[2] Cranfield Soil and Agrifood Institute, Cranfield University, Cranfield MK43 0AL, UK; r.sakrabani@cranfield.ac.uk
[3] Chemical Engineering Department, Visvesvaraya National Institute of Technology, South Ambazari Road, Nagpur 440010, Maharashtra, India; mandavgane@gmail.com
\* Correspondence: k.patchigolla@cranfield.ac.uk

**Abstract:** The choice of activating agent for the thermochemical production of high-grade activated carbon (AC) from agricultural residues and wastes, such as feedstock, requires innovative methods. Overcoming energy losses, and using the best techniques to minimise secondary contamination and improve adsorptivity, are critical. Here, we review the importance and influence of activating agents on agricultural waste: how they react and compare conventional and microwave processes. In particular, adsorbent pore characteristics, surface chemistry interactions and production modes were compared with traditional methods. It was concluded that there are no best activating agents; rather, each agent reacts uniquely with a precursor, and the optimum choice depends on the target adsorbent. Natural chemicals can also be as effective as inorganic activating agents, and offer the advantages that they are usually safe, and readily available. The use of a microwave, as an innovative pyrolysis approach, can enhance the activation process within a duration of 1–4 h and temperature of 500–1200 °C, after which the yield and efficiency decline rapidly due to molecular breakdown. This study also examines the biomass milling process requirements; the influence of the dielectric properties, along with the effect of washing; and experimental setup challenges. The microwave setup system, biomass feed rate, product delivery, inert gas flow rate, reactor design and recovery lines are all important factors in the microwave activation process, and contribute to the overall efficiency of AC preparation. However, a major issue is a lack of large-scale industrial demonstration units for microwave technology.

**Keywords:** activated carbon; activating agent; adsorption; agricultural waste; biomass; chemical activation; kinetic model; microwave activation; waste utilisation

## 1. Introduction

Activated carbon (AC), is a carbonaceous solid derived from coal or biomass via thermal or thermochemical processes. ACs are typified by their well-developed pore morphology, remarkably high surface area, electron-conducting amphoteric tendencies and high adsorptive capacity [1]. AC is used widely in various applications beyond adsorption to treat various industrial effluents. Currently, the global market for AC is worth several billion dollars annually [2].

AC offers one of the best means for dealing with water and air pollution issues, which pose major health risks [3]. Such problems arise from effluent discharges from a wide array of industries, including the textile, brewing, chemical and food processing ones, and are due to inappropriate disposal of

domestic and industrial wastes, and thermal processes, such as coal combustion [4]. Advances in water treatment processes are dependent on the removal of impurities, heavy metals in particular [5]. Generally, the use of AC is one of the most acceptable means of waste water treatment, because of the wide range of production methods and target system applications for adsorbates [6]. Most often, the application of the adsorbate determines the production method and activating agent [7]. AC is also useful as a battery component, such as in the production of capacitors [8]. AC can also be useful for treating the inhalation of poisonous gases and in cosmetic applications, like teeth whitening [9]. Finally, ACs are also effective for drug overdose, hemoperfusion, removal of endogenous and exogenous toxins in uraemia and in the dressing of suppurating wounds to decrease odour [10].

AC can be synthesized from feedstocks, such as petroleum residue, coal (in particular lignite), agricultural residues, wood and a range of biomass materials. Crop residue production is ubiquitous where agriculture is practiced and its production is more than 3107 Mt/a for 17 varieties of cereals and 25 varieties of legumes, and 3758 Mt/a for 27 food crops at the commercial level [11]. Based on data from 227 countries, global production of agricultural residues for cereals was estimated to be about $3.7 \times 10^{12}$ t/a [11]. Waste management and disposal costs could be reduced by utilizing such materials in AC production [12]. The abundance of agricultural waste generated globally remains a significant environmental issue [13] and AC production offers the prospect of using it as a renewable carbon source to produce highly porous AC. Another option is biochar production from agricultural waste, which is produced at much lower temperatures (250–500 °C) and typically has an average Brunauer–Emmett–Teller surface area ($S_{BET}$) of 200 $m^2$/g, while for AC, $S_{BET}$ typically exceeds 700 $m^2$/g.

According to global AC data, the market was estimated at $2007 million in 2012 and is expected to reach $5305 million by 2020. Here, the compound annual growth rate was estimated to be about 13.3% from 2014 to 2020 [14]. The chemical characteristics of AC are defined by the surface groups and chemical bonding of heteroatoms. The surface functional groups influence characteristics such as polarisation intensity, hydrophobicity, acidity and adsorption properties. The optimum activation temperature is also subject to various factors, such as physiochemical traits of the biomass, oxygen content, electrical and catalytic properties and the pollutant targeted [15].

There are several methods of producing AC. However, the use of microwaves for AC production is attracting more research interest due to its numerous advantages compared to conventional heating techniques. These advantages include: uniform volumetric heating, internal heating, higher, selective heating, good regulation of the process and indirect contact with the heat source [16]. Microwaves specifically function with electromagnetic radiation, within the frequency scope of 300 MHz to 300 GHz [17]. Domestic and industrial microwave units sometimes operate in the same frequency range; however, most are designed for 2.45 GHz frequency, which is equivalent to 122 mm and $1.02 \times 10^{-5}$ eV for wavelength and energy respectively [18]. The fundamental physics of microwaves' applications with respect to material treatment or AC production are dependent on the compounds adsorbed and the carbon matrix [19]. Maxwell equations derived from appropriate boundary conditions can be used to describe the microwave process [20].

$$P = \left[ (\sigma + \omega\epsilon'')E^2 + \omega\,\mu''\,H^2 \right]. \tag{1}$$

The power density available for adsorption P is given in Equation (1), and is measured in W/$m^3$. Here, E represents the electric field vector; the vector H is the magnetic field; and σ and μ are: electro conductivity and magnetic permeability, respectively. The dielectric factor and angular velocity are denoted with $\epsilon''$ and ω, respectively. Microwave processes can control the heating and materials being heated internally and uniformly, and rapidly with high efficiency to pyrolyse large particles [21].

This study reviews agricultural waste as an AC precursor regarding its activation conditions and chemical activating agents, with a substantial emphasis on the characteristic properties of AC [22] and adsorption application targets [23]. While there are reviews on this subject [24,25], none to date have

considered the pre-processing challenges of parent materials, and only provide limited insight into kinetic and adsorption parameters and the important issue of scale-up.

This review attempts to evaluate multiple activating agent effects and make generic comparisons and assessments of multiple agricultural wastes, while outlining current process challenges. In particular, this study considers the effectiveness of some of the chemical activating agents used for AC production under various production conditions and also looks at the effects on the product characteristics, of surface functional groups and of overall adsorption efficiency for several applications. Comparative analyses of production methods are also examined to suggest more appropriate pathways. Finally, this review also concentrates on the behaviour and chemical properties of precursors relative to several treatments, production conditions, comparative analyses of modes of production and their challenges.

## 2. Fuel Characterisation for AC Production

Agricultural waste can be thermally and chemically treated to generate a wide range of valuable products, such as biofuels, bio-oils, bio-gases and bio-solids [25]. It can also be considered an energy storage medium [26]. For such wastes, it is essential to allow for the wide variation in chemical contents. The proximate, ultimate and lignocellulose contents of such materials through biochemical analysis are shown in Table 1. The three main structural components: hemicelluloses, cellulose and lignin [27] are produced in varying amounts depending on whether they are produced by thermochemical [28] or biochemical conversion processes, and this affects the properties of the AC.

Agricultural waste is abundant, renewable and available at low cost [29]. Its use is potentially environmentally friendly [30]. However, there are some caveats; Abbas and Ahmed [31] showed that leaves are not good for AC, due to their low carbon content, high volume to weight ratio and ash content. Research has also shown that in chemical activation, the activating agent digests amorphous lignin better than it does the biomass cellulose, so it is important to understand the characteristics of any material used [32].

Lignocellulosic biomass contains 10%–30% lignin, and materials such as coir, have about 45% lignin. Lignin is a natural insoluble polymer, and contains aldehydic and carbonyl groups, which make it highly polarised [33]. Typically, during carbonisation and activation, the initial volatiles removed come from cellulose and hemicellulose [34]. Several lignocellulosic materials possess special characteristics for AC production [35] and there are also several simple and low-cost methods for obtaining AC from biomass, using acids, bases and salts [36]. The cross-linking between the lignocellulosic components determines the suitability of biomass material for AC, and the morphology of the cells is also a determinant [37]. The energy requirement is subject to the characteristic properties of the feedstock. The structure of biomass materials, the physiognomies and the chemical behaviour of each component affect the reactivity and have potential involvement in processing AC [32].

Cellulose and hemicellulose are similar, with the main difference being the number of saccharide units, the latter having fewer saccharide units than the former [23]. Cellulose has a straight chain structure and is classified as an insoluble polysaccharide made up of monomers of beta-1, 4-glycosidic bonds between glucose. By contrast, hemicellulose is a cross-linked polymer with some sugars which play vital roles in AC production, such as glucose, mannose, galacturonic acid, xylose, arabinose, O-methyl-glucuronic acid and galactose, which are all are soluble in water.

The thermal decomposition of hemicellulose takes place above 200 °C, after which cellulose's decomposition follows; more specifically, in the temperature range of 250–400 °C with the release of CO and $CO_2$ from the glycopyranose rings [23]. Lignin is hydrophobic and inhibits water penetration, unlike polysaccharide polymers. The molecular relationship expressed in Figure 1 [38], are of ether and β-1,4-Glycosidic bonds. Therefore, these linkages must be broken thermally or chemically to establish a reactive activated carbon.

**Figure 1.** A structural model of lignocellulosic components of biomass.

**Table 1.** Analytical characterisation and composition of agricultural waste biomass.

| Agricultural Waste | Proximate Analysis (% w/w) | | | Ultimate Analysis (% w/w) | | | | | Lignocellulosic Composition (% w/w) | | | Reference |
|---|---|---|---|---|---|---|---|---|---|---|---|---|
| | Moisture | Ash | Volatiles | C | H | N | S | O | Cellulose | Hemicellulose | Lignin | |
| Almond stone | 11.05 | 0.76 | 77.32 | 48.76 | 7.52 | 0.48 | 0.56 | 43.68 | 21.70 | 27.70 | 36.10 | [39] |
| Bamboo | 15.30 | 1.76 | 70.12 | 34.40 | 4.61 | 0.22 | 0.07 | - | 26 | 15 | 21 | [40] |
| Banana peel | 11.56 | 9.28 | 88.02 | 35.65 | 6.19 | 1.60 | 20.75 | 45.94 | - | - | - | [41] |
| Cassava peel | 14 | 4.50 | 59.40 | 59.31 | 9.78 | 2.06 | 0.11 | 28.74 | 37.90 | 23.90 | 7.50 | [42] |
| Coconut shell | 8.21 | 0.80 | 77.82 | 49.62 | 7.31 | 0.22 | 0.10 | 42.75 | 14.00 | 32.00 | 46.0 | [43] |
| Cotton stalks | 6.00 | 6.30 | 70.50 | 43.60 | 5.80 | 0.80 | 0.00 | 49.80 | 80-95 | 5-20 | - | [44] |
| Durian shell | 11.27 | 4.84 | - | 39.30 | 5.90 | 1.00 | 0.06 | 53.74 | 60.45 | 13.09 | 15.45 | [4] |
| Grape stalk | 8.86 | 3.15 | 96.80 | 34.40 | 0.438 | 1.11 | 0.087 | 63.96 | - | - | - | [45] |
| EFB | 15.01 | 4.48 | 82.98 | 43.89 | 5.33 | 0.52 | 0.10 | 54.32 | 42.00 | 18.90 | 11.70 | [46] |
| Oil Palm MF | 11.10 | 7.90 | 84.03 | 42.20 | 5.21 | 2.21 | 0.14 | 42.34 | 42 | 22 | 14 | [46] |
| PKS | 7.96 | 1.10 | 72.47 | 50.01 | 6.90 | 1.90 | 0.03 | 41 | 20.80 | 22.70 | 50.70 | [4] |
| Olive stone | 10.40 | 1.40 | 74.40 | 44.80 | 6.00 | 0.10 | 0.01 | 49.09 | 30.80 | 17.10 | 32.60 | [47] |
| Orange peel | - | 2.15 | 77.93 | 40.28 | 6.12 | 1.08 | 0.06 | 52.46 | - | - | - | [48] |
| Peanut shell | 7.98 | 12.80 | 79.10 | 41.52 | 7.43 | 2.12 | 0.60 | 27.96 | - | - | - | [49] |
| Rice husk | 6.34 | 16.70 | 67.50 | 36.52 | 4.82 | 0.86 | - | 41.10 | 30.42 | 28.03 | 36.02 | [50] |
| Sugarcane BG | 8.61 | 4.05 | 86.02 | 47.30 | 6.20 | 0.27 | - | 44.15 | 42.16 | 36.00 | 19.30 | [50] |
| Walnut shell | 8.73 | 1.27 | 77.42 | 49.30 | 5.82 | 44.49 | 0 | - | 40.10 | 20.70 | 18.20 | [47] |
| Waste tea | 5.80 | 4.29 | - | 52.72 | 6.34 | 2.61 | 0.18 | 38.15 | 17.50 | 41.30 | 41.20 | [51] |

C: carbon, H: hydrogen, N: nitrogen, S: sulphur, O: oxygen, EFB: empty fruit bunch, MF: mesocarp fibre, PKS: palm kernel shell, BG: bagasse.

### 3. Modes of Activation

The preparation of AC typically involves two major steps: pyrolysis or carbonisation of the precursor, and activation. Carbonisation produces a stable structure, with an elementary and partially-developed pore structure [52], which must be enlarged properly by physical or chemical activation. The development stages are illustrated in Figure 2. In addition, physicochemical activation is also a potential form of activation [16]. Figure 2 illustrates the pore development of empty fruit bunch (EFB) AC, represented with A, B and C, following thermal treatment at 350, 500 and 600 °C, respectively [53]. The pore development is better defined with increased temperature, and modification and treatment methods for raw biomass are discussed by Pathak et al. [54].

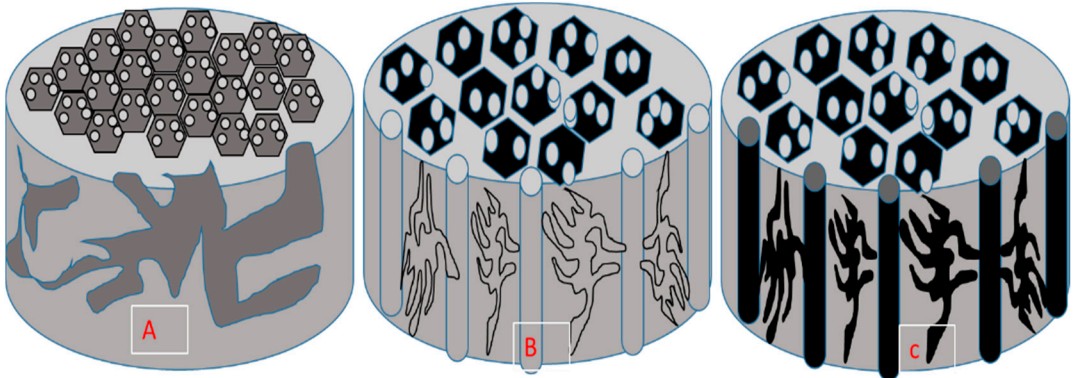

**Figure 2.** Activation temperature's effect on activate carbon's (AC) pore development and morphology (**A**): biomass structure; (**B**): partially developed char structure; (**C**): well-defined porous AC structure.

Physical activation occurs over the dual stages of carbonisation of the precursor in a reactor with a steady flow of an inert gas, and then over a successive activation process of the char product with $CO_2$. Steam, air or their mixtures or other fluid activating agents at 800–1100 °C are also used [55]. However, excessive temperatures of, say, 1200 °C and above, lead to low carbon yields, collapse of the pore structures and ash generation [56], while devolatilising processes encourage pore formation and further enlarge any micropores created [57]. Physical activation is environmentally safe, but the speed and the temperature requirements are problematic. Appropriate activation temperature and duration are important to ensure adequate porous development and the creation of functional groups.

The physicochemical activation process involves both physical and chemical methods [58], where the raw precursor or char is permeated with an activating agent; then, heated in a chamber with an oxidant flow [59]. This method is often employed when the activating agent used in activation is unable to be effectively removed through washing, and might otherwise lead to pore clogging [33]. This process is expensive due to the elevated temperature requirements, and the need for a two-step process, extended process time and a generally low-percentage AC yield [59]. There are various modifications and treatments, including acidic treatment, basic treatment, impregnation and microwave process treatment. Other routes include: ozone, plasma treatments and biological modification—which is a developing technology and will not be discussed here further [6]. Although conventional heating is still an effective method of AC production, there are basic challenges that affect the process, such as the high cost of the heating, extended heating duration and thermal energy flow from the outer layer to the core of a biomass material. Chemical treatment of agricultural waste and other biomass materials can also improve the overall efficiency of AC and provide opportunities for target applicability due to the potential to use several activating agents for the process [60]. Chemical activation processes are summarised in Table 2 with the key advantages and disadvantages presented.

## 3.1. Chemical Activation

A series of pre-activation steps is required to achieve an adequate and quick production process. Biomass materials must be free from soil and any other impurities by washing with deionized water, and then dried at a temperature between 65 and105°C [59]. The biomass selected often undergoes milling to reduce the particle size [61], enabling uniform and quick carbonisation by lowering the thermal gradient. The biomass physical properties influence the milling process. The chemical activation process can be achieved in several ways, as outlined below:

The chemical activation method is accomplished by the infusion, mixing and permeation of a solution, usually a dehydrating chemical with the ability to induce and accelerate material decomposition by pyrolysis, while inhibiting the creation of semi-solid volatile substances before activation. Activating agents, such as bases; acids; and salts, such as $K_2CO_3$ and $ZnCl_2$ are impregnated into the biomass by mixing and stirring [62]. Although this is a single-step technique, in some cases, the precursor may first be carbonised to produce char [63,64] The impregnated char is then heated in the presence of nitrogen [65]. The temperature requirement used in chemical activation range of 400–800 °C, which is lower than the average temperature for the physical activation process [64] and is applicable to lignocellulosic materials. Here, the raw precursor can be mixed and permeated with an activating agent directly before carbonisation [66]. Typically, the biomass material is added to the activating agent at an optimised ratio, and mixed, before undergoing thermal treatment. Each activating agent performs differently with different precursors, dependent on its structure and the activation conditions. It is also possible to use a two-step activation process by impregnating biochar with activating agent and then subjecting it to a second heat process, under the conditions outlined in Table 2 [63].

**Table 2.** An overview of the three modes for the chemical activation process.

|  | One-Step Conventional | Two-Step Conventional | Microwave Process |
|---|---|---|---|
| **Temperature or power requirements** | 400–1200 °C | 400–800 °C then 400–1200 °C | 300–1000 W |
| **Heating duration** | 1–3 h | 3–6 h | 5–30 min |
| **Average yield** | 30–50% | 30–40% | >40% |
| **Risk of system corrosion** | High | High | Low |
| **Product efficiency** | Low | High | High |
| **Flow process** | Continuous feed in and out | Batch | Batch |

### 3.1.1. Chemical Activation

The one-step process system for AC production requires impregnation before heat treatment under a controlled environment, and this is known as direct activation. A wide thermal range between the hot exterior and the interior risks encourages significant combustion of the biomass. The microwave system can be used for this process and will shorten the process time compared to a two-step process [67].

### 3.1.2. Two-Step Activation Process

Two-step process systems for AC production involve carbonisation and then activation [68]. This process enhances the carbon content and forms a char with an initial porosity. It requires an extended time and consumes more energy. It can also be used for non-uniform particles so that a milling stage is not required in pre-processing. This process is common for microwave activation, but in the production of large quantities of AC, the transfer and reloading lead to loss of materials and increase production cost. Oginni et al. [69] reported that the two-step process does not result in more effective AC.

### 3.1.3. Microwave Activation Process

The high energy requirement and time involved in conventional heating can make the activation process expensive [70], making microwave heating attractive [71]. Some biomass types are good microwave absorbers, and hence, are interesting for possible use at the commercial scale [72]. Microwave-induced pyrolysis is more efficient, especially when a material with higher microwave absorbance is mixed with metal oxides [73]. Microwave pyrolysis reduces self-gasification during the activation process, unlike conventional pyrolysis, due to the high heating levels that are involved in conventional approaches [74].

Compared to conventional heating methods, the microwave method of activation has additional benefits, including uniform volumetric heating, regulation of the heating process, a high heating rate and ensuring that the biomass has indirect contact with the heat source. Thus, the microwave AC production technique has been suggested as a prospective method to replace the conventional thermal process [16]. In some cases, the activation process is planned according to the availability of resources and the heat source. Activation duration, though, depends on the type of precursor but can be affected strongly by the particle size and heat distribution mechanism. Carbon yield depends on the overall management of the sample, its temperature, its washing and handling techniques.

### 3.2. A Comparison of Activation by Microwave and the Conventional Process

Microwave-assisted activation produces AC with wider micropores and mesopores than the conventional products. The pore size, surface area and adsorption efficiency in both conventional and microwave activation processes are affected by activation temperature, dwell time, heating rate, type of activating agent, activation agent-feedstock ratio, particle size, inert gas flow rate, microwave power and weight of sample [75].

Figure 3 shows that heat flows from the external regions in conventional heating and from the interior for microwave heating [76]. This is the distinguishing factor between conventional and microwave heating. Direction of heat flow relative to the activation environment influences the molecular movement, volatile discharge and heat distribution. Conventional heating exhibits an extended temperature gradient, while thermal energy flows from the surface to the central portion of the biomass. However, it can be minimised by slow heating. An isothermal holding system may also be employed [77]. A thermal gradient results in the obstruction of the passage for volatiles through the pores, leading to the development of partial or undefined pores [49,77].

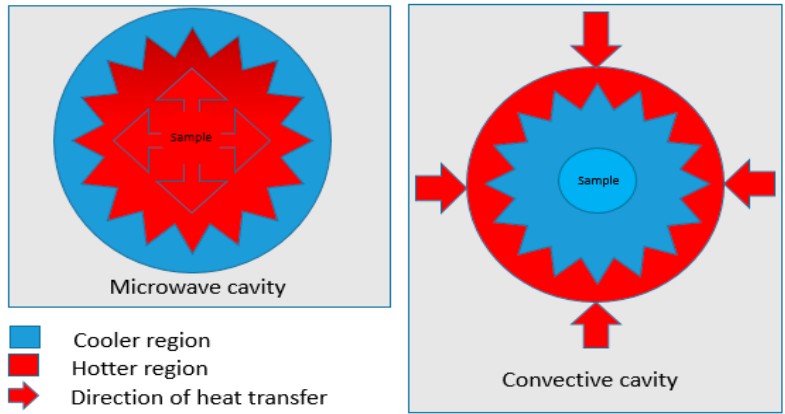

**Figure 3.** Examples of conventional and microwave heating effects.

Microwave AC is characterised by wider micropores and mesopores than conventional activated AC, but it comes with process challenges [77,78]. Xin-hui et al. [64] studied activation approaches for AC production from Jatropha hulls with steam as the activation agent, and showed the carbon yields for microwave and conventional methods were 16.6% and 13.3%, respectively. With $CO_2$ as an

activating agent, the yield for conventional method was 18%, and it was 36.6% for the corresponding microwave method. Foo and Hameed [78] compared EFBAC produced through a conventional furnace and microwave and reported $S_{BET}$ and $V_{total}$ values of 255.77 $m^2$/g and 0.14 $cm^3$/g, and 807.54 $m^2$/g and 0.45 $cm^3$/g, respectively. A general overview of the two approaches is shown in Table 3 [74,79].

**Table 3.** Comparison of the differences between various processing parameters for microwave and conventional activation methods.

| Framework | Microwave Activation | Conventional Activation | Observation |
|---|---|---|---|
| Treatment time | Shorter process time | Several hours and days | Shorter time reduces process risks |
| Heating process | Internal and volumetric heating | Surface heating Non-uniform heating | Thermal gradient is eliminated by microwave heating |
| Mode of heat transfer | Energetic coupling Coupling at molecular level | Conduction and convection Superficial and wall heating | Uniform heating is easily achieved on coupling at molecular level |
| Gas/Energy consumption | Due to short treatment time, it is low | High | Microwave process saves energy |
| Equipment size | Small | Large | Developing a large size microwave is expensive and complex |
| Preparation conditions | 300–700 W 5–15 min | 400–1200 °C 1–3 h | Same for both except when two-step activation process is required |
| AC characteristics | Higher surface area | High surface area | Choice may depend on application and process requirements |
| Complexity | High | Low | Conventional equipment is easy to build. Equipment repair is easier with conventional approaches |

## 4. Activating Agent Effect

### 4.1. The Effects of Activating Agents on Renewable and Non-Renewable Precursors

Activating agents play major roles in the activation process, and the various compounds used react differently depending on the biomass type and the temperatures employed.

Several non-renewable precursors (petroleum tar pitch and coal) have been used for AC production. When coke is impregnated with KOH, the porosity decreases as the ratio varies from 4:1 to 2:1, and the results obtained are in the following ranges: $S_{BET}$, 1800–1200 $m^2$/g; $V_{micro}$ 0.71–0.48 $cm^3$/g; and mean pore size, 1.12–0.76 nm [79]. AC from bituminous coal can be produced by chemical activation with KOH and Borax decahydrate, but the resulting AC produced has an irregular pore structure.

Similarly, paracetamol, phenol and salicylic acid can used for coal-based ACs [79], while nitric acid is used for the activation of pitch, resulting in an AC with a $S_{BET}$ of 1401 $m^2$/g and an adsorption capacity of 3.51 mol/kg, and is employed for the adsorption of $CO_2$ [80]. The microwave process for coal, produced an AC with a 1048 mg/g adsorption capacity for methyl blue (MB) and values of 1770 $m^2$/g and 0.99 $cm^3$/g, for $S_{BET}$ and $V_{total}$, respectively, under production conditions of a 1:3 impregnation ratio for 12 min at 700 W [81].

Gao et al. [82] showed tar residues permeated with $H_3PO_4$ were activated at 850 °C, resulting in the following: adsorption capacity, $V_{total}$ and $V_{micro}$ values of 793 mg/g, 0.286 $cm^3$/g and 0.255 $cm^3$/g, respectively. In addition, the mixture of coal and carbon monolith for AC production had a $S_{BET}$ of 1044 $m^2$/g [83].

The reactivity of chemical activating agents on biomass materials under high thermal conditions creates an intermolecular reaction that gives rise to efficient AC production. Surface textural analysis

shows that the increase in the ratio of activating agents is proportional to the efficiency until the optimal range is reached, and this creates AC with higher pore volume on activation. There are clear morphological differences depending on the activating agent; thus, the $K_2CO_3$-impregnated sample forms columnar and hexagonal shapes. Köseoılu and Akmil-Başa, showed that $ZnCl_2$-impregnated AC forms an asymmetrical and irregular surface morphology [81], and they found that $K_2CO_3$ produced AC with a higher $S_{BET}$ and a well-defined porosity; however, the implication of such studies is not clear, as results are strongly dependent on technical activation conditions and impregnation parameters.

Microwave treatment has the potential to eliminate some volatiles, starch molecules and saturated and acidic oxygen-linking functional groups; therefore, making the AC basic; thus, promoting reactivity [84]. The type of bonding on the carbon layer is influenced by type of feedstock, microwave power, impregnation ratio and radiation time. Other variables, such as inert gas flow rate, can influence AC characteristics. These have a substantial impact on the adsorption capacity and other applications of AC [85].

The adsorption capacity is attributable to the effectiveness and reactivity of ions and ionic exchange in the carbon structure. The increase in charge density is proportional to the adsorption capacity [83]. At certain ranges of microwave power, pores can be blocked by deposits formed from volatile substances [86]. Raw materials, precursor particle size, activation process and mode of heat treatment also influence the surface chemical and functional groups of ACs [87]. Foo and Hameed [88] identified the removal of loosely-bound atoms and switching with heteroatoms, which are naturally basic, within the molecular layers. For the microwave activation process, the power requirement is critical due to possible collapse of the pore structure beyond the optimum heat requirement [89].

Activating agents possess great capacity in the microwave activation process. Biomass materials, due to poor dielectric properties, heat quickly with higher thermal penetration when impregnated with chemical agents [90]. For example, AC prepared from orange peel when all processing parameters are constant, showed substantial changes. However, any activation effect is insignificant, and minimal adsorption occurs at microwave power below 200 W. A study by Foo and Hameed [91] showed that adsorption capacity increases directly proportionally to the increase in microwave power.

Deng et al. [92] explored the influence of microwave power on coconut shell-based AC. Therein, $K_2CO_3$ and KOH were used for impregnation and the resulting adsorption capacity was similar at same microwave power [93]. In some cases, impregnation ratio influences AC properties and is used in the optimum range of 0.5–2 wt.% [94]. A further challenge in AC production is incomplete activation, which results in poor adsorption caused by partial elimination of the acidic functional groups, and the dominance of saturated molecule and incomplete removal of volatiles [84]. Fourier-transform infrared spectroscopy (FTIR) evaluation shows the presence of individual bonds and clearly defines the difference between carbonisation and activation. It can also be used to determine the optimum range of activation at different parameters. The peaks are indicative of the reactive sites and functional groups responsible for adsorption [95].

## 4.2. Acidic Treatment of Agricultural Residues

The sensitivity of cellulose linkage to acidic hydrolysis is due to the presence of glycoside bonds and it varies relative to type of acid, the concentration and the reactive temperature within the amorphous region. It is homogenous with strong acids and heterogeneous with weak acids. The kinetics of the hydrolysis reaction are also governed by decrystallisation of microcrystalline cellulose. Hydrolysation, which results in the elimination of xylan, helps in the separation of complex structures. Most acids are able to transform crystalline cellulose to amorphous, while HCl can remove lignin to enable symmetric pore creation. The cell walls of plants, where cellulose and hemicellulose have polymeric bonds, are easily broken by acids, and that results in delignification.

Introducing nitric acid into AC production was successful in improving the physical properties of AC. $HNO_3$ successfully removed some contaminants attached to the surface of AC, such as Fe, Si, K and Al. The surface of AC pores was clearly seen to have unbroken cavities when treated [96] and

showed an improved hydrophilic character [6] Comparing HCl and HNO$_3$ under the same conditions for palm kernel shell (PKS) activation produced: S$_{BET}$ values of 164.2 and 325.4 m$^2$/g; V$_{total}$ values 0.10 and 0.19 cm$^3$/g; and pore diameters 9.50 and 8.00 nm, respectively. H$_3$PO$_4$ is corrosive to both metals and tissues, undergoing rapid polymerisation with epoxides and polymerisable compounds, but it can separate lignin and cellulose from lignocellulose [97]. H$_3$PO$_4$ breaks down cellulose and hemicellulose polymers in biomass to form monomeric sugars. It can also react directly with lignin under any conditions, and cellulose dissolves in it. H$_3$PO$_4$ has been used extensively in the activation of AC, whose adsorption isotherm indicates multilayer filling, with significant mesopore formation. H$_3$PO$_4$, used for AC production from olive stone, has a good S$_{BET}$ of about 1218 m$^2$/g and a good pore volume of 0.6 cm$^3$/g [98]. Örkün et al. [99] studied the impact of concentration variations of H$_3$PO$_4$ on some precursors and considered it effective for the development of pores and producing a significant improvement of the morphology, surface chemistry and size distribution. A significant contribution to microporosity and mesoporosity was observed for grape seed activation in the 1:1 to 3:1 ratio range, especially in the pore size arrangement.

### 4.3. Base Treatment of Agricultural Residues

Alkaline can cause the degradation of glucose and cause the branching of carbohydrates, such as 4-O-methylglucuronic acid. They are reactive to lignin and the solvation of ether linkages and hydroxyl groups. This is useful in AC production due to delignification. The dissolution of lignin causes the removal of xylan. The use of KOH results in the breakage of longer fibres and exfoliation [100]. Improved porosity as a result of activation with KOH can be attributed to potassium interaction and the stretching of carbon layers. KOH is more effective at creating micropores due to the evidence of an interaction between layers, while NaOH is extremely active for disordered carbon materials; however, both activating agents require temperature above 800 °C. Figure 4 shows how NaOH' interacts with biomass to breakdown the long chains.

**Figure 4.** The interactions between layers and NaOH.

The use of KOH, in the temperature range of 230–650 °C, results in a high char yield. The oxidation of biomass with KOH creates oxygen-containing surface functional groups, which may not develop properly at temperatures below 600 °C [101]. KOH is highly-effective in biomass; sugar cane activated by KOH has an S$_{BET}$ of 2300 m$^2$/g with high methyl blue adsorption [102]. During the KOH activation process, surface species, such as alkaloids, as well as molecular species like K$_2$CO$_3$ and K$_2$O, are formed [101]. NaOH impregnation of corncob resulted in high S$_{BET}$ between 2318 and 2474 m$^2$/g. When using the optimum ratio of 1:2 to 1:1, the mesopore volumes of the AC were increased from 21% to 58% [103]. AC perpared from date palm showed competitive characteristics of 1282.49 m$^2$/g, 0.66 cm$^3$/g and 20.73 nm—S$_{BET}$, V$_{total}$ and average pore diameter, respectively. The adsorption was

determined to be lowest at pH 3 (104.88 mg/g), increasing to 106.4 mg/g at pH 7. AC properties vary indirectly with temperature, and a maximum of 612.1 mg/g at 30 °C for relative adsorption capacity was observed with the AC from guava seed, which showed an effectiveness for amoxicillin adsorption with a $S_{BET}$ of 2573 m$^2$/g and an average pore diameter of 1.96 nm [1].

*4.4. Salts Treatment of Agricultural Residues*

The hydrolysis of hemicelluloses is strongly affected by temperature and pH; a near neutral pH lowers the activation process; however, ethers and carbon-carbon bonds are relatively stable. Some salts can catalyse the ether linkages in the lignin. At a high temperature, some salts undergo the Leidenfrost effect, causing steam explosion; thus, liberating soluble phenolics in the liquid form. The hydrophilicity of lignin is increased due the bond breakages at that stage. The increase in the reaction time or temperature results in the degradation of oligomers and monomers. That could also influence degradation to produce glycoaldehyde dimer, D-fructose, 1,3-dihydroxyacetone dimer, anhydroglucose, 5-HMF and furfural. Salt can also act as a catalyst to influence the gasification rate of cellulose at the low temperature of about 400 °C.

$ZnCl_2$ is commonly used as an activating agent in AC production, especially with biomass to obtain a high-reactivity and high-surface-area AC [104]. $ZnCl_2$ as an activating agent acts as a dehydrating agent when impregnated in biomass, resulting in hydrolysis reactions because of intermolecular exchange and molecular migration which would cause structural alteration due to weight loss and discharge of volatiles. Hydrocarbon and oxygenated organic compounds are not affected by $ZnCl_2$, thereby providing the AC with a skeleton with developed pores. Şahin et al. [105] studied AC preparation from Elaeagnus angustifolia seeds using $ZnCl_2$ at 500 °C, a 1:1.5 impregnation ratio for 48 h and activation for a 1 h duration, resulting in an $S_{BET}$ of 1836 m$^2$/g. Potato peel biomass was studied for AC production with $H_3PO_4$, KOH and $ZnCl_2$. The activation was done at 400 °C, resulting in: $S_{BET}$ 1642 and 1489 m$^2$/g, and $V_{total}$ 0.96 and 0.93 cm$^3$/g, for KOH and $ZnCl_2$, respectively [106]. $K_2CO_3$ has been proven effective for several agricultural waste precursors. For sisal, for example, the resultant AC had an $S_{BET}$ of 1038 m$^2$/g, $V_{total}$ of 0.49 cm$^3$/g and was confirmed to be effective and appropriate for the purification of solutions contaminated by ibuprofen and paracetamol from the liquid phase. Studies on bamboo and pinewood showed a higher $S_{BET}$ when impregnated with $K_2CO_3$ with microwave activation than for any other method. Foo and Hameed [86] compared $K_2CO_3$ and KOH for pineapple peel for 6 minutes in a microwave; the results showed ACs with $S_{BET}$ values of 680 and 1006 m$^2$/g, respectively. Furthermore, comparison using grapeseed as the precursor, activated at 800 °C with $K_2CO_3$ and KOH, yielded ACs with $S_{BET}$ values of 1238 m$^2$/g and 1222 m$^2$/g, respectively. The average pore diameter of both remained the same, at 1.7 nm [107].

$FeCl_2$ and $FeCl_3$ as activating agents can boost the intermolecular interactions between activating agent and biomass precursor. There are limited investigations on the use of $FeCl_2$ and $FeCl_3$ for biomass AC production [108]. At the temperature range of 400–1000 °C, $FeCl_3$ as an activating agent proved itself effective on Tara gum with a resultant $S_{BET}$ of 1680 m$^2$/g and a $V_{total}$ of 1 cm$^3$/g [109]. Under the same conditions for Arundo donax Linn activation, results were: $S_{BET}$ 760 and 927 m$^2$/g; $V_{total}$ 0.466 and 0.509 cm$^3$/g; and average pore diameters (nm) 2.451 and 2.11, for $FeCl_2$ and $FeCl_3$, respectively [110]. Another comparative study with $ZnCl_2$, $FeCl_3$ and $MgCl_2$ for waste coffee biomass resulted in $S_{BET}$ values of 123, 977 and 846 m$^2$/g, respectively [111]. The comparative study of AC synthesis from coffee ground waste with $ZnCl_2$ and $FeCl_3$ resulted in $S_{BET}$ values of 977 and 846 m$^2$/g, respectively. Further study with the same precursor using $MgCl_2$ gave a value of 123 m$^2$/g [111], and the same trend was observed for the date stone [112]. $MgCl_2$ also proved to be effective in activation of Tara gum with $S_{BET}$ of 1680 m$^2$/g [113]. Figure 5 and Table 4 clearly show the disparity of different activating agents compared alongside the production methods. However, this comparative table shows that most activating agents under optimum production conditions have different effects on different precursors.

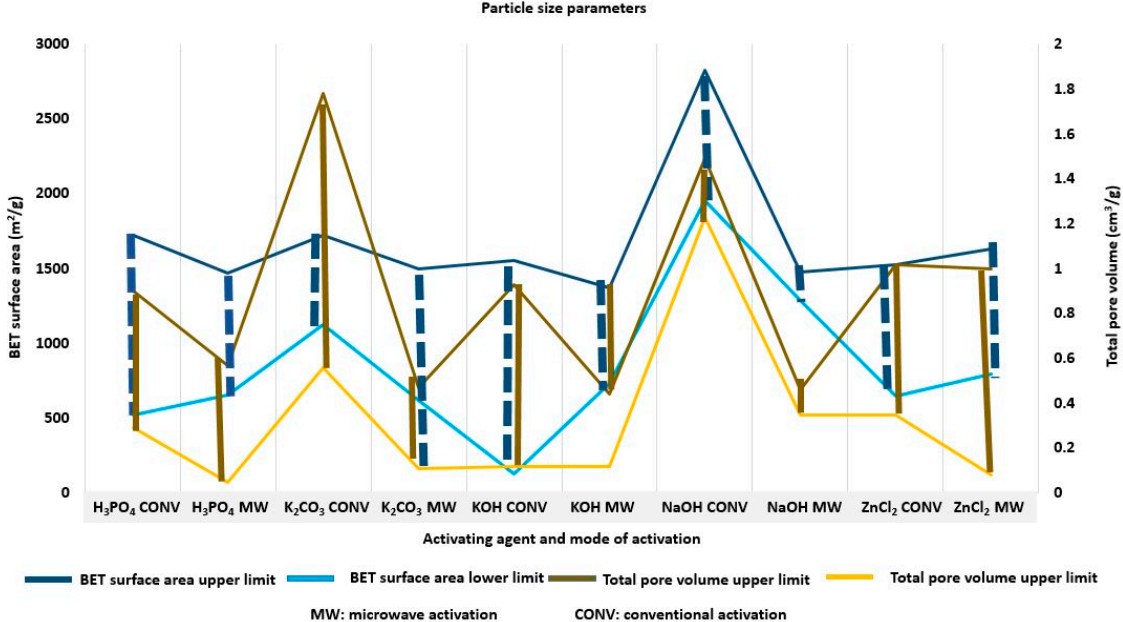

**Figure 5.** $S_{BET}$ and total pore volume ranges of some activating agents on agricultural waste.

The fact that primary characteristics of AC depend on the pyrolytic conditions and activating agents, mean that the hydrogen and oxygen contents decrease with increases in carbonisation and activation temperature. The aromaticity of carbon and morphology increases as the hydrogen/carbon ratio decreases. The relationship between temperature and surface area is not linear due to other factors that contribute to the characteristics of AC. The impregnation time and activating agent are additional determinants of these characteristics because of their direct influence on the pore sizes, and thus, release of volatiles. However, an extended impregnation period can result in the gasification of pore walls. Therefore, the burn-off as a factor should be adequately considered. Figure 5 and Table 4 show that acids, bases and $ZnCl_2$ are more effective for activation [110–144].

**Table 4.** The effects of activating chemical on the precursors.

| Precursor | Time (hr) | T(°C)/MW(W) | I.R | $S_{BET}$ (m$^2$/g) | $V_{total}$ (cm$^3$/g) | $V_{meso}$ (cm$^3$/g) | $V_{micro}$ (cm$^3$/g) | $D_P$ (nm) | Yield (%) | Reference |
|---|---|---|---|---|---|---|---|---|---|---|
| **Result of H$_3$PO$_4$ activation of conventional process** | | | | | | | | | | |
| Cotton stalk | 2 | 500 | 1.5 | 1720 | 0.890 | | 0.710 | | | [114] |
| Peach stones | 2 | 500 | 0.4 | 1393 | 0.689 | 0.055 | 0.634 | 0.99 | 41.8 | [115] |
| Bamboo | 1 | 600 | 1.5 | 1335 | 0.625 | 0.140 | 0.485 | 1.87 | | [116] |
| Maize tassel | 1 | 500 | 1.4 | 1263 | 1.592 | | | | | [56] |
| Palm shell | 0.5 | 425 | 2 | 1109 | 0.903 | | | 3.20 | | [117] |
| Durian shell | 0.4 | 500 | | 1021 | 0.350 | | 0.210 | | 63 | [118] |
| Cotton cake | 1.5 | 450 | 2 | 584 | 0.298 | 0.075 | 0.223 | 2.04 | 29.8 | [119] |
| Rice straw | 2 | 450 | 1 | 522 | 0.550 | 0.370 | 0.180 | 4.21 | 51.9 | [120] |
| **Result of H$_3$PO$_4$ activation of microwave process** | | | | | | | | | | |
| Palm shell | 0.28 | 800 | 2 | 1473 | | | | | | [121] |
| Lotus stalk | 0.25 | 700 | 2 | 1434 | 0.307 | 1.030 | 1.337 | | | [84] |
| Bamboo | 0.5 | 350 | 1 | 1432 | 0.503 | 0.193 | 0.696 | | 48 | [116] |
| Waste tea | 0.5 | 350 | 3 | 1157 | 0.573 | 0.256 | 0.829 | 35 | | [122] |
| Cotton stalk | 8 | 400 | 0.5 | 652 | 0.057 | 0.419 | 0.476 | 2.92 | | [123] |
| **Result of K$_2$CO$_3$ activation of conventional process** | | | | | | | | | | |
| Waste tea | | 900 | 1 | 1722 | 0.583 | 0.039 | 0.554 | 2.0 | 22.7 | [51] |
| Rice husk | 1.5 | 1000 | 1.5 | 1713 | 1.785 | 1.070 | 0.715 | 4.16 | | [94] |
| Mangosteen shell | 2 | 900 | 1 | 1123 | 0.560 | 0.110 | 0.450 | 1.98 | 20.7 | [124] |

**Table 4.** *Cont.*

| Precursor | Time (hr) | T(°C)/MW(W) | I.R | $S_{BET}$ $(m^2/g)$ | $V_{total}$ $(cm^3/g)$ | $V_{meso}$ $(cm^3/g)$ | $V_{micro}$ $(cm^3/g)$ | $D_p$ (nm) | Yield (%) | Reference |
|---|---|---|---|---|---|---|---|---|---|---|
| **Result of $K_2CO_3$ activation of microwave process** | | | | | | | | | | |
| Wood saw dust | 0.1 | 600 | 1.3 | 1496 | 0.470 | 0.394 | 0.864 | 2.30 | 80.0 | [125] |
| Rice husk | 0.11 | 600 | 1.3 | 1165 | 0.330 | 0.450 | 0.780 | 2.68 | | [94] |
| Orange peel | 0.1 | 600 | 1.3 | 1104 | 0.247 | 0.368 | 0.615 | 2.22 | 80.9 | [91] |
| Pineapple peel | 0.1 | 600 | 0.8 | 680 | 0.280 | 0.170 | 0.450 | 2.59 | | [86] |
| Cotton stalk | 0.13 | 660 | 0.8 | 621 | 0.110 | 0.270 | 0.380 | 2.43 | | [126] |
| **Result of KOH activation of conventional process** | | | | | | | | | | |
| Rice straw | 1 | 800 | 4 | 1554 | 0.930 | 0.340 | 0.500 | 1.14 | 13.5 | [120] |
| Bamboo | 3 | 800 | 2 | 1533 | 0.491 | | | | | [40] |
| Cocoa pod husk | 1 | 800 | 1 | 490 | 0.240 | | | 2 | 13.5 | [127] |
| PKS | 0.75 | 800 | 1 | 127 | 0.120 | | 0.110 | | 43.4 | [128] |
| **Result of KOH activation of microwave process** | | | | | | | | | | |
| Empty FB | 0.11 | 600 | 1 | 1372 | 0.440 | 0.320 | 0.760 | 2.20 | 73.7 | [78] |
| Mesocarp Fibre | 0.1 | 600 | 0.8 | 1223 | 0.420 | 0.300 | 0.720 | 2.35 | 32.0 | [129] |
| Pineapple peel | 0.1 | 600 | 0.8 | 1006 | 0.280 | 0.310 | 0.590 | 23.44 | | [86] |
| Palm Shell | 0.16 | 600 | 0.6 | 895 | | | 0.491 | 2.19 | | [128] |
| Rice husk | 0.11 | 600 | 1.3 | 752 | 0.260 | 0.380 | 0.640 | 3.41 | | [94] |
| Coconut husk | 0.11 | 600 | 1.3 | 752 | 0.260 | 0.380 | 0.640 | 3.41 | | [130] |
| Cotton stalk | 0.16 | 680 | 0.6 | 729 | 0.120 | 0.260 | 0.380 | 2.08 | | [126] |
| **Result of NaOH activation of conventional process** | | | | | | | | | | |
| Coconut shell | 1.5 | 700 | 3 | 2825 | 1.498 | 0.355 | 1.143 | 2.27 | 18.8 | [131] |
| Rice husk | 0.05 | 850 | 2 | 1958 | 1.230 | 0.573 | 0.550 | 1.25 | | [132] |

**Table 4.** *Cont.*

| Precursor | Time (hr) | T(°C)/MW(W) | I.R | $S_{BET}$ (m²/g) | $V_{total}$ (cm³/g) | $V_{meso}$ (cm³/g) | $V_{micro}$ (cm³/g) | $D_p$ (nm) | Yield (%) | Reference |
|---|---|---|---|---|---|---|---|---|---|---|
| **Result of NaOH activation of microwave process** | | | | | | | | | | |
| **Durian shell** | 0.1 | 600 | 1.5 | 1475 | 0.467 | 0.374 | 0.841 | 2.28 | 80.0 | [133] |
| **Langsat bunch** | 0.1 | 600 | 1.3 | 1293 | 0.449 | 0.303 | 0752 | 2.32 | 81.3 | [134] |
| **Jackfruit peel** | 0.11 | 600 | 1.5 | 1286 | 0.356 | 0.408 | 0.764 | 2.37 | 80.8 | [135] |
| **Result of ZnCl₂ activation of conventional process** | | | | | | | | | | |
| **Tea seed shell** | 1 | 500 | 1 | 1530 | 0.783 | 0.184 | 0.599 | 2.05 | 44.1 | [136] |
| **Oceania palm** | 2 | 600 | 4.5 | 1483 | 1.022 | 0.456 | 0.494 | 2.76 | 40.0 | [137] |
| **Waste apricot** | 1 | 500 | 1 | 1060 | 0.790 | 0.640 | 0.150 | 2.98 | | [81] |
| **Date stone** | 1.2 | 500 | 2 | 1045 | 0.641 | 0.129 | 0.512 | 2.45 | 40.4 | [138] |
| **Rice husk** | | | | 927 | 0.560 | | | 0.80 | | [139] |
| **Palm Shell** | | 900 | | 926 | 0.480 | | 0.470 | 2.05 | | [140] |
| **Wood apple shell** | | | 2 | 794 | 0.470 | | | | | [141] |
| **Hazelnut shell** | | | | 647 | 0.350 | | | 34.0 | | [142] |
| **Result of ZnCl₂ activation of microwave process** | | | | | | | | | | |
| **Peanut shell** | 0.3 | 600 | 1.4 | 1634 | | | | | | [143] |
| **Rice husk** | 0.3 | 600 | 1.4 | 1527 | | | 2.070 | 5.99 | | [143] |
| **Cotton stalk** | 0.15 | 560 | 1.6 | 794 | 0.083 | 0.547 | 0.630 | 3.20 | 37.5 | [144] |

IR: impregnation ratio, $V_{total}$: total volume, $V_{meso}$: mesopore volume, $V_{micro}$: micropore volume, $D_p$: pore diameter, nm: nanometre, FB: fruit bunch.

## 5. Characterisation of AC

### 5.1. Surface Chemistry Mechanisms and Morphology

AC porosity, pore volume, size, surface area and other structural characteristics are relative to adsorption capacity. FTIR analysis identifies the functional groups and chemical on the edges and planes of ACs, which are very important in AC characterisation and in prediction of adsorption performance. The surface chemistry and functional groups of AC are reliant on the precursor, activating agent, activation process and conditions. Franca et al. [83] identified oxygen, nitrogen and hydrogen, as important atoms on the surface of AC, because of their ability to bond with other elements [1].

In surface chemistry analysis, metal ions are bound by electrostatic forces into the permeable networks at acidic pH, as most effective adsorptions occur below pH 7. This is because of the strong role of the oxygenated acidic surfaces. Some adsorption at higher pH is assisted by chelating and complex formation [24]. Carboxyl groups also contribute to heavy metal adsorption by biomass adsorbents while phenolic groups are associated with the formation of complexes with heavy metal pollutants [145], and these could be direct determinants of the effectiveness of AC. While FTIR analysis provides important information, monoatomic speciesdo not possess infrared spectra and this makes complex mixtures difficult to analyse [146].

A comparison of the FTIR spectra of bamboo feedstock, traditional and conventionally activated AC and microwave-assisted activated AC [116] shows the effect of the $H_3PO_4$-induced microwave activation of bamboo's spectral pattern and shows that thermal activation with a furnace and microwave have distinctive differences. Pathak and Mandavgane examined the FTIR frequencies and the resultant effects of activation with citric acid on banana peels, showing that there are major differences dependent on the method of activation and the activating agent [147].

Huang et al. [84] demonstrated that $H_3PO_4$ breaks bonds and dissociates aliphatic and aromatic molecules, especially in biomasses, encouraging the removal of many volatile substrates, and initiating partial aromatisation and carbonisation for a defined AC. A study investigating pineapple peel impregnated with KOH activation showed FTIR spectra of the AC as major peaks within these ranges: 3230, 2360, 2040, 1420, 1050 and 830 $cm^{-1}$, indicating the location of N–H and C≡C, which are highly unstable; C=C—moderately unstable; and C–N, C–O and N–O functionalities, respectively, which indicates AC's capacity to permit attachment [86]. KOH and $K_2CO_3$-induced AC were both enhanced in relation to its char and developed better when activated by microwave irradiation [94]. The same activation process when used on a cotton stalk as a precursor showed identical absorbance bands. However, there was substantial variance in the spectra of $ZnCl_2$-activated AC. Sharp bands of 1630 and 1600 $cm^{-1}$ are distinctive and are absent in KOH and $K_2CO_3$ ACs [126]. The surface chemistry of NaOH-induced AC from pomelo skin was observed to have an FTIR spectrum within 3413 and 3247 $cm^{-1}$ correlated to the amine groups, and the band at 2980 $cm^{-1}$ indicated the presence of an alkane group [88]. For PKS, AC was produced through microwave and conventional methods with $ZnCl_2$; the O–H stretching vibration receded, validating the effectiveness of the two processes. The significant effect and reduction of hydrogen bonding spectral bands show that the $ZnCl_2$ typically dehydrates the sample from the point of impregnation.

### 5.2. Adsorption and Kinetics Mechanism

Pyrolysis takes place under inert conditions; this causes the release of volatile substances that form tar at elevated temperatures, depending on the type and quantity of material. Current kinetic models are inadequate for biomass pyrolysis, specifically in terms of the prediction of pyrolysis rate. Molecular modelling can potentially identify the pathways for cellulose and lignin to production of biofuels and AC. Models presume that kinetic constraints, such as the frequency factor and activation energy, are interdependent and contingent on both conversion mechanism and the extent of heating. Kinetic models presume a series of decompositions, for $n^{th}$ order reactions. By contrast, the distributed activation energy model (DAEM) describes the pyrolysis reaction itself, and postulates that many

decomposition, n$^{th}$ order reactions occur simultaneously, with distributed activation energies [148]. There are also double exponential terms which can be used to describe the distributed activation energy model for isothermal pyrolysis. In order to find the accurate approximation, an asymptotic methodology is implemented [148]. The amount of tar depends on the temperature and duration of carbonisation. Feed rate also affects tar deposition, due to prevention of the escape of volatiles. Furthermore, the type of activating agent is also critical in determining the rate of tar transformation into gas.

Adsorption can be classified as physisorption or chemisorption. Physisorption uses van der Waals forces, whose effects rapidly decrease with increasing temperature [149], while chemisorption results in the formation of chemical bonds between the adsorbate and adsorbent [150]. Several factors and variables influence adsorption processes of pollutants with AC. Consequently, AC adsorbs through van der Waals, hydrogen bonding and electrostatic charge [151]. The functional groups of the adsorbent are primarily relative to the production process. Skouteris et al., [152] showed that, in some types of adsorption processes, such as in liquid-phase, the adsorption efficiency and potency of AC are strongly dependent on the physical structure of the AC, the pore assemblage and the functional groups with respect to the parent materials, and also showed that molecular weight and polarity of adsorbent are influential. Other secondary properties of the solutions that are potentially important include the adsorption thermodynamics, pH, molecular diffusion, concentration of adsorbate and ionic energy [153].

Adsorption capacity is generally determined by the percentage of the pollutant removed, calculated as in Equations (2) and (3) [154]:

$$q_t = \left(\frac{C_0 - C_t}{m}\right) V ,\tag{2}$$

$$R = \left(\frac{C_0 - C_t}{C_0}\right) 100\% ,\tag{3}$$

where, $q_t$ (mg/g) symbolises the adsorption capacity, $C_0$ represents, initial pollutant concentration at start time and $C_t$ (mg/cm$^3$) represents pollutant concentration at time (t). V (cm$^3$) denotes the volume of the adsorption medium and m (g), the adsorbent weight.

The adsorption performances of ACs derived from agricultural wastes are shown in Table 5. Unfortunately, the literature usually provides insufficient information on the adsorption parameters. However, AC activated by $H_3PO_4$ is very effective in the removal of contaminants and adsorption, taking place at 25–30 °C. Adsorption occurs mostly in the pH range of 4–7. Salt activators have a wide range of pH for adsorption, which also depends on concentration of the solution.

Adsorption isotherms are series of measurements relative to adsorbed and non-adsorbed amounts in a single process. The gas and solute adsorptions are functions of pressure and are dependent on chemical structure. Isotherm models are necessary for AC when a batch system for wastewater treatment is designed. This evaluates the suitability of the adsorption system, also showing the system to be endothermic or exothermic [69]. The thermodynamic assumptions provide a guide to the adsorption mechanism, adsorbent affinity and the fundamental approaches. The models are illustrated in Table 6 and consider several assumptions and parameters for plotting adsorption isotherms.

**Table 5.** Adsorption capacities and parameters of chemically-activated agricultural wastes based on activating agent.

| Adsorbent | Activator | Activation Temp /MW Power (W) | IR | Adsorbate | $S_{BET}$ m²/g | Adsorption Capacity (mg/g) | CR | Contact Time (min) | Contact Temp °C | Contact pH | Reference |
|---|---|---|---|---|---|---|---|---|---|---|---|
| Durian shell | $H_3PO_4$ | 500 °C | 1:0.30 | Toluene | 1404 | 874 | | 2 h | 25 | | [155] |
| Waste tea | $H_3PO_4$ | | | MB | 1398 | 288.34 | | | | | [156] |
| Sugarcane bagasse | $H_3PO_4$ | 500 °C | 3:1 | Pb | 320 | 170.90 | 6 | 30 min | 25 | 4 | [157] |
| Cotton stalk | $H_3PO_4$ | 500 °C | 3:2 | Pb(II) | 1570 | 119 | | | 25 | 4.4 | [158] |
| Olive stone | $H_3PO_4$ | 600 °C | 1:1.5 | Cd | 1565 | 24.83 | | | | 5 | [159] |
| Coconut husk | KOH | 700 °C | | MB | 1356 | 418.15 | | | | | [130] |
| Banana peel | KOH | 800 °C | | MB | 2086 | 385.12 | | 60 | | 6 | [160] |
| EFB (char) | KOH | 360 W | 1:0.75 | MB | 807 | 344.80 | 50 | | | 4 | [78] |
| Date stone | KOH | 600 W | 1:1.75 | MB | 856 | 316.10 | | | | | [161] |
| Coconut shell | KOH | 700 °C | 1.5:1 | Benzene | 478 | 212.77 | | | 30 | | [162] |
| Orange peel | NaOH | | | Reactive Blue 19 | | 166.60 | 100 | 24 h | 30 | 4 | [163] |
| Coconut shell | KOH | 800 °C | 1:2 | Pb | 1135 | 151.52 | 4 | | | 5 | [164] |
| Peanut shell | KOH | | | Cr(VI) | 96 | 16.26 | 40 | 7 h | 25 | 4 | [165] |
| Tea seed shell | $ZnCl_2$ | 500 °C | 1:1 | MB | 1531 | 342.70 | | | | 6.3 | [136] |
| Pineapple waste | $ZnCl_2$ | 500 °C | 1:1 | MB | 915 | 288.34 | | 12 h | | | [166] |
| Corn cob | $K_2CO_3$ | 600 W | 1:0.75 | MB | 765 | 275.32 | | | | | [167] |
| Cotton stalk | $ZnCl_2$ | 560 W | 1:1.6 | MB | | 193.50 | 4 | 2 h | | 7 | [144] |
| Pistachio nut shell | $ZnCl_2$ | | | Hg | 1492 | 24.78 | | 100 min | 25 | 6 | [168] |
| Hazelnut shell | $ZnCl_2$ | 700 °C | | Pb | 1067 | 13.05 | 200 | 10 min | | 5.7 | [169] |
| Olive stone | $ZnCl_2$ | 650 °C | | Cd | 790 | 1.85 | 50 | 90 min | 25 | 9 | [170] |

IR: impregnation ratio, CR: concentration range, MW: Microwave, W: watt, MB: methyl blue.

**Table 6.** Adsorption isotherm models descriptions and challenges.

| Isotherm [171] | Nonlinear Form [171] | Linear Form | Plot | Nomenclature | Description/Assumptions |
|---|---|---|---|---|---|
| Langmuir | $q_e = \frac{Q_0 b c_e}{1 + b c_e}$ | $\frac{q_e}{C_e} = b Q_0 - b q_e$ | $\frac{q_e}{C_e}$ vs $q_e$ | $q_e$=equilibrium sorption capacity for metal $C_e$=equilibrium solute concentration in solution b=Langmuir constant | -The simplest isotherm -Assumes that adsorption is limited to monolayer -There is insignificant interaction and change in the sorption process -The model is identical to energy being adsorbed |
| Freundlich | $q_e = K_f C_e^{1/n}$ | $\log q_e = \log K_F + \frac{1}{n} \log C_e$ | $\log q_e$ vs $\log C_e$ | $K_F$= equilibrium constant for relative adsorption n =adsorption constant indicative of intensity | -Relative to multi-layer adsorption -widely applied in organic compounds and highly interactive substance |
| Sip's isotherm | $q_e = \frac{K_s C_e^\beta s}{1 + a_s C_e^\beta s}$ | $\beta In(C_e) = -In\left(\frac{K_s}{q_e}\right) + In(a_s)$ | $In\left(\frac{K_s}{q_e}\right)$ vs $In(C_e)$ | $\beta$ = Sip's constant | -combination of Langmuir and Freundlich -It is governed by pH, temperature and concentration changes |
| Temkin | $q_e = \frac{RT}{bT} In A_T C_e$ | $q_e = \frac{RT}{b_T} In A_T + \left(\frac{RT}{b_T}\right) In C_e$ | $q_e$ vs $In C_e$ | $A_T$ and $b_T$ =Tempkin constants | -Takes care of adsorbate-adsorbent interaction -Used for predicting the gas-phase equilibrium |
| Khan | $q_e = \frac{q_s b_k C_e}{(1 + b_k C_e) a_k}$ | | | $b_k$ and $a_k$ are model constant and model exponent | -General adsorption model for adsorbate from pure dilute equation solutions |
| Redlich-Peterson | $q_e = \frac{K_g C_e}{1 + a_g C_e^\beta}$ | $In\left(K_R \frac{C_e}{q_e} - 1\right) = g In(C_e) + In(a_R)$ | $In\left(K_R \frac{C_e}{q_e}\right)$ vs $In(C_e)$ | KR, aR, and $\beta$ = Redlich–Peterson parameters. $\beta$ takes values between 0 and 1. For $\beta$=1 the model can be correlated to the Langmuir form | - It is anti-ideal monolayer adsorption -Has a linear relative dependence on concentration of solution |
| Radke-Prausnitz | $q_e = \frac{a_{gp} r_g C_e^{\beta g}}{a_{gp} + r_g C_e^{\beta e - 1}}$ | | | The outlook of surface diffusion Relative role of pore volume. aR and rR = model constants | -Preferred in most systems at low adsorbate concentration Good for wide range of adsorbate concentration |
| Frenkel-Halsey-Hill | $In\left(\frac{C_e}{C_s}\right) = -\frac{\alpha}{RT}\left(\frac{q_s}{q_e^d}\right)^r$ | | | Where d, $\alpha$ and r = sign of the interlayer spacing (m), | |
| Toth | $Q_e = Q_{max} \frac{\frac{b_T C_e}{\left[1 + (b_1 C_e)^{1/\eta\tau}\right]^{\eta\tau}}}{}$ | | | bT and nT = constants. if nT=1, Toth isotherm can be transformed to Langmuir isotherm equation | -Reduces error in experimental data -Predicts equilibrium data -Applicable in the modelling of several multilayer and heterogeneous adsorption systems |
| Dubinin-Radushkevich | $q_e = (q_e)\exp\left(-k_d \varepsilon^2\right)$ | $In(q_e) = In(q_s) - K_{ad}\varepsilon^2$ | $In(q_e)$ vs $\varepsilon^2$ | $k_d$ = isotherm constant. $\varepsilon$ = RT ln(1+1/$C_e$) R, T and Ce = gas constant CeTemperature (K) | |

## 6. Production Process Challenges

### 6.1. Washing and Milling

Some biomass wastes, such as oil palm residue (OPR), have a high percentage of tramp material, including soil, while oil palm processing waste (OPW) does not, and instead has a high oil content. Hence, the need for washing can vary significantly. A high water requirement raises the production cost and hot water may be required to remove the oil attached to the biomass precursor. If the oil is not properly removed, it produces copious amounts of smoke during carbonisation, and if the temperature is not high enough (say below 400 °C), it forms tars within the biomass material, preventing the escape of volatiles, thereby resulting in improper development of pore structure, which requires a high inert gas flow rate to effectively displace volatile materials.

OPW is produced mostly in the form of large particles and lumps; therefore, size reduction is crucial for utilisation in AC production. This is done by milling Figure 6 [172], a high-energy operation [173]. Particle size is one of the crucial determinants in the thermochemical route; physical condition of biomass notwithstanding, the method influences activation rate and other production parameters [174]. There is also difficulty in milling mesocarp fibre (MF) and EFB, rather than palm kernel shell (PKS), due to low brittleness. Conventional hammer mills typically have a 250 mm diameter rotor at a rated power of about 20 kW [175], but there may be a need to use swinging hammers operating at high velocity [176]. Biomass waste can be crushed when dry, sieved and graded into different uniform sizes of less than 2 mm using a milling machine [126], to achieve uniform distribution of heat. The mechanical properties of different parts of OPW vary and it is important to minimise dust production. To minimise loss of biomass, a dust collector is installed, which aids in throughput flow and overcomes air flow resistance [177]. Pre-treatments, such as drying and sieving, occasionally play a role in determining energy consumption. Establishing uniformity of particle sizes is largely contingent on their moisture content and other physical properties. Brittleness, hardness and elasticity also affect the energy requirements in milling. Phillip et al. [178], studied the mechanical and physical influence of PKS, and showed that the aggregate crushing value is 5.3% and that of the flakiness index is 63.2%, indicating that PKS milling behaviour is acceptable.

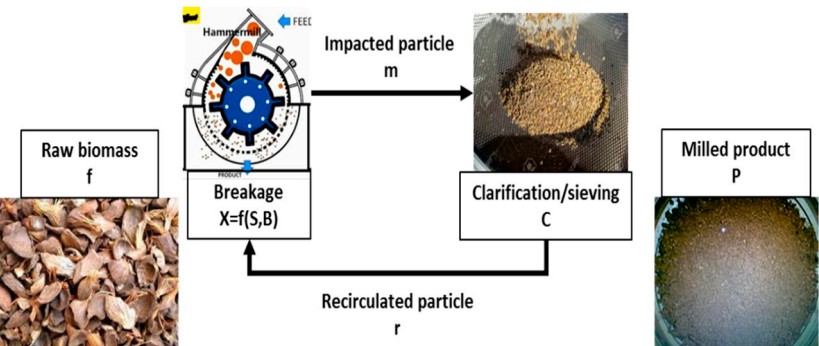

**Figure 6.** Hammer milling process for biomass.

The efficiency of a hammer milling operation is determined by the nature of the biomass, and thus, milling tendencies vary widely. Several physical and mechanical properties are important, e.g., material breakage, as the fracture reaction is contingent on impact frequency, energy, moisture content and biomass properties [172]. Vogel and Peukert [179] expressed the phenomenon of breakage as in Equation (4):

$$S = 1 - exp\left\{-f_{mat} x k\left(W_{m,kin} - W_{m,min}\right)\right\}. \tag{4}$$

X represents the initial particle size, and k and $W_{m,ki}$ are the impact frequency and specific kinetic energy of the impact, respectively. $W_{m,min}$ is the threshold energy. The material properties in terms of its resistance and minimum specific energy are relevant to determine fracture and breakage factor

which define the milling rate [180]. The energy requirement increases with reducing particle size [181]. However, it should be noted that the physical properties influence feeder design, biomass flow rate and reactor performance [182]. For each batch of AC production, the biomass should be of uniform particle size. Higher density also translates to lower tensile strain. The energy requirement for the milling process contributes to about 15%–30% of the overall energy requirement for AC production [181].

## 6.2. Microwave Power and Radiation Duration

The energy requirement is dependent on the type of reactor and the effect of power is relative to the duration of the process. The reaction temperature is maintained after the heating stage, as power remains constant. Evaluating the influence of microwave power on yield, time is a primary factor. Hoseinzadeh Hesas et al. [95] assessed the influence time of the microwave radiation relative to the surface consistency of AC, in order to optimise power output to achieve the appropriate heating rate and process temperature. High radiation power of above 1000 W for an extended time may result in total carbonisation with the adverse effect of reducing surface area and porosity [183]. For activation, the power range of 100–200 W is inadequate to create complete release of volatile substances, thereby resulting in improper development of pores with negligible influence on the adsorption capacity. At about 300–1000 W, the adsorption efficiency increases and pore development is better defined. However, with a higher thermal radiation there is greater loss in weight beyond the optimum range and collapse of the pore structures; thus, there needs to be a balance between activation duration and microwave power.

Microwave radiation duration is another important factor and strongly affects the resulting characteristics of the activated material. Thus, using the microwave heating method for a period of 5 min, the $S_{BET}$ initially increases to a maximum value of 2996 m$^2$/g at 20 min, and then hardly changed with the increase in the activation duration for the char of Mesocarbon microbeads [184].

Adsorption can be improved by an increase in radiation time. For example, with pineapple as a precursor, a 4 min radiation time difference increased adsorption from 206 to 283 mg/g, which then decreased after 6 min as carbon yield decreased. Further increase in process time beyond the optimum always leads to the formation of hotspots and collapse of pore structures [86]. A typical duration is 5–20 min for char, but if raw biomass is impregnated with activation agent, an appropriate time will be about 15 min. Moisture content could also create imbalance in the prediction table for process duration.

## 6.3. Activation Interaction and Temperature

The inter-molecular interaction within the biomass precursor under an electromagnetic field involves the release of an enormous amount of energy relative to the quantity of material being processed. The process outcome is dependent on the dielectric factors of the precursor [185]. Reaction during activation is highly influenced by the type of activating agent. Each biomass material reacts differently to different activating agents and the products differ. Palm kernel shell (PKS) and other oil palm wastes (OPW) were used in biochar production by microwave process, at different radiation levels. The biochar yields were 38, 35 and 33 wt.%, for 500, 600, 700 W respectively [186,187]. Typically, more volatiles are released as microwave power is increased. In conventional systems, the major carbon losses occur during milling, sieving and washing. The losses during carbonisation can be minimised by the right choice of carbonisation temperature range. Mushtaq et al. [188] have explored the effects of sample variability, particle size, moisture content, type of activating agent and inert gas flow rate on the pyrolysis and activating process and concluded that each type of biomass must be individually characterised.

The external surface temperature can be measured with infrared pyrometer; however, not during the actual processing itself. Another challenge of microwave radiation is the generation of hot spots because of the non-vegetative impurities in the precursor [79,94]. The thermal energy and temperature value of the sample core can be up to 500 °C higher than the external temperature [95]. Nonetheless, microwave power of 450 W is considered to be the lowest possible power required to cause significant

changes in the in the biomass's structure for production of an efficient AC; below that, AC cannot be effectively produced [84]. In order to reduce AC process time, a novel method of impregnating OPW char with CuO receptors resulted in the rapid increase of sample thermal energy, although its use resulted in the generation of widely varying pore structures [189].

Unfortunately, due to limited choices of measuring technique, sample temperature measurement was often ignored in previous works on microwave-assisted reactions [189]. There are three options for the measurement of temperature in microwaves operation: shielded thermocouples; IR-sensors, which measure infrared radiation as passive sensors; and a fibre optics sensor, which measures temperature up to 400 °C using an external transducer based on the wavelength modulation principle. Shielded thermocouple measurement is a cost-effective option [184]. However, it is inappropriate in the presence of non-polar solvents. In AC production occurring above 300 °C, thermocouples are not recommended [189]. The temperature of the sample during MW treatment can also be measured by using an infrared pyrometer [184]. In general, there is difficulty in ensuring good temperature regulation, though the use of a water jacket is possible [190]. However, such an approach will not work for any large-scale process, given the thermal inertia inherent in such a system. A built-in-sensor on the reactor and further research on the appropriate reactor material could help to overcome such challenges.

### 6.4. Tar Deposition

The constituents and characteristics of tars depend on the type of fuel and thermal process. The formation of tar is common; however, at a high temperature, primary tars vapours can be cracked. This situation can be influenced by reactor.

The primary tars contain substances such as dimethoxy phenol, trimethoxybenzene and hydroxy methoxy benzoic acid, and these can be transformed into xylene phenol and toluene. There are tertiary tars also contain materials such as benzene, naphthalene, acenaphthylene, phenalene and pyrene. The presence of tar compounds such as 1-methylnaphthalene, which has a boiling point of 244.8 °C, are problematic, as they can emit acrid smoke during decomposition.

The presence of tar in a thermal process creates complex problems, especially in the utilisation downstream. During AC production, AC can re-adsorb it. Tar can be removed by decomposition or adsorption by activated carbon. In the process of AC production, tar causes reduction in the performance of AC [191]. Phenol and cresol available in biomass can be degraded by microbial agents such as yeast [192] in tar removal.

A study by Abu El-Rub et al., [193] showed that tar can be removed by thermal cracking and a catalytic processes. It is necessary to address the influence of tars due to their carcinogenic character, ability to form coke and plugging which could disrupt the system. However, some catalysts can also affect the characteristics of AC. Finally, the use of alkali–metal–base for the removal of tar can also result in particle agglomeration at a high temperature.

### 6.5. Equipment Selection and Mode

The laboratory scale microwave typically only produces small quantities of material and 10–20 g of AC is typical. Pyrex glass reactors and Teflon reactors generally perform well in the laboratory but may not be suitable for scaling up. Ceramic/solid sintering materials also seem suitable for microwave processing but require further study. Evaluating AC production by means of laboratory-scale microwave systems is inexpensive, but cannot provide appropriate understanding of the molecular interactions and transformational kinetics in AC production at scale [187], and there is an urgent need for results from appropriately sized equipment.

There are advantages related to using a concentrated stream of energy when utilising unimode microwave radiation. However, most practical systems will use multimode radiation in a thermochemical process [189]. Multimode is relatively cheap and can be designed to deal with large sample sizes. However, unimode is better in terms of controllability [194]. The energy release of unimode radiation depends on reactor material, size, waveguide channel, the properties precursor and

other contents of the sample. Generally, these devices have magnetrons directed through a waveguide and an electromagnetic field [189]. Operation is also influenced by the mixing of biomass and activating agents, the cooling method, control heating rate, the seal and microwave leakage rate. To overcome these technical challenges and reduce overall process time, a continuous process should be used, since it eliminates wastage during transfer and mechanical washing, and ensures uniform production, as most operations are automated.

*6.6. Dielectric Properties*

The dielectric properties of any material are primary factors in the transformation of electromagnetic energy into heat for AC production. Most biomass has poor microwave absorbing properties [195]. Abubakar et al. [196] explored the influence of dielectric properties of OPW and several studies have explored the dielectric properties of agricultural waste and other biomass materials.

The relative dielectric constant, dielectric loss factor ($\varepsilon_r$) and tangent loss for OPW and corresponding biochar, at varying frequencies (0.2–10 GHz) and at ambient conditions (room temperature of about 25 °C) are given in Table 7, for a range of materials. The dielectric constant for most materials decreases with increasing frequency [195], while the loss factor varies in the opposite direction. Most biomass materials are only weak absorbers. The average loss tangent of biomass feedstock is less than 0.1. The addition of activating agents improves the microwave absorbability [197]. In some biomass materials, absorbability increases rapidly at elevated temperatures and this can result in process challenges due to thermal runaway [198]. The poor understanding of the relative influence of electromagnetic energy on combined and mixed biomass on the microwave heating remains a challenge in obtaining homogeneous products [20]. High moisture content in a material can affect its dielectric properties and result in a high microwave treatment efficiency [197].

**Table 7.** Relationship between penetration depth of biomass and various microwave frequencies [196,199,200].

| Material | Frequency (GHz) | Penetration Depth (cm) | Permittivity ($\varepsilon'_r$) | Tangent Loss (Tanδ) |
|---|---|---|---|---|
| PKS | 0.9–5.8 | 5.5–36 | 2.7 | 0.13 |
| MF | 2.5–5.8 | 10.2–24.8 | 2.0 | 0.08 |
| EFB | - | - | 6.4 | 0.3 |
| Water | 2.54 | 1–4 | - | - |
| Paper | 2.54 | 20–60 | - | - |
| Wood | 2.54 | 8–350 | 2.3 | 0.11 |
| Natural Rubber | 2.54 | 15–350 | 2.1 | 0.003 |

During carbonisation and activation, the delivery of products helps to release volatiles. When the volatile products are not desired, the issue of how to exhaust them arises in the design setup. In most laboratory tests and analyses of microwave-induced activation, modified domestic systems are used for the work, so are not relevant for industrial applications. There are also a few issues with the use of stirrers in microwave systems. Small particles are capable of elutriating with any vapour generated, especially at higher stirring speeds, affecting the carbon yield; therefore, a stirrer should not be installed close to the volatiles delivery line and size should be designed to reduce arcing which is seen as flashes or sparks [201].

## 7. Scale of Microwave for Use in Preparing AC

The technique of the microwave method to produce AC depends first and foremost on the scale of potential units. In 2010, Leonelli and Mason published an article claiming that microwave and ultrasonic processing have good prospects for industry, but the focus was primarily on the food

industry [202]. Progress in the food area seems to have continued, with microwaves being used in particular for lower-temperature applications, such as drying, although interestingly, even for this type of application, there are issues with the formation of hot spots [203]. Generally, what has been offered is still relatively small, with levels of 6 L/h being suggested as adequite [204]. However, there does seem to be some recent progress and a current paper suggested that microwave units were available able to process flow rates of up to 750 kg/h for drying [205]. There is also at least one company offering a microwave reactor with a capacity of 7 m$^3$, which seems to be sufficient for this type of application [206]. Interestingly, a recent patent for AC preparation from nuts, gave the technology a technological readiness level of 5 [207]. Thus, it is clear that, at least to date, there are no full-scale demonstrations of microwave technology producing AC at an industrial scale, and it is there that future research is essential if this technology is to be meaningfully used to make commercial quantities of AC in the near future [208].

## 8. Summary

The chemical activation approach for AC production is widely applied because of its ability to produce AC with a wide range of activating agents under a myriad of processing conditions. The choice of activating agent is contingent on the adsorption target and precursor. Each activating agent has a unique impact on the characteristics and applications of the resultant AC. However, due to the challenges of AC production from biomass, there is a need to develop techniques to achieve industrial-scale production of AC. Microwave systems accelerate the activation process. Bases, acids and ZnCl$_2$ are generally considered to be particularly effective. Unfortunately, natural chemicals are rarely used for activation despite their potential to reduce secondary contamination which arises when materials like ZnCl$_2$ and inorganic acids are used, due to the difficulty of effectively removing them by washing. It is clear that microwave treatment is potentially better than conventional approaches, especially in terms of energy saving and process duration; however, to date there are no commercial application of such technology. Despite the existence of larger and larger microwave systems, there remains an urgent need to demonstrate this technology at a commercial or near-commercial scale.

**Author Contributions:** K.S.U., concept development and preparation of the original manuscript; K.P., design and formulation of the research question and supervision; R.S., arrangement of the manuscript and supervision; E.A., modified and edited the manuscript; S.M., data collection for comparative study. All authors have read and adopted the manuscript.

**Funding:** This research was approved and funded by PETROLEUM TECHNOLOGY DEVELOPMENT FUND (PTDF), Nigeria, under the sponsorship code (PTDF/ED/PHD/UKS/1007/16).

**Conflicts of Interest:** The authors declare no conflict of interest.

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
