# Peer review of "A Review of Chemicals to Produce Activated Carbon from Agricultural Waste Biomass"

_sustainability, doi:10.3390/su11226204_

Round 1
Reviewer 1 Report
The review is dedicated to researches of relations between pyrolysis conditions (heating, reagents) and characteristics of activated carbon.
The paper is organized and written quite well. Nevertheless, I have some questions.
1) It can be concluded from fig. 5 that microwave heating leads to worse quality of activated carbon: surface area and adsorption capacity are less than at convective heating. So question arises: does microwave heating allow to enhance activation process? What is weighted profit of microwave systems in terms of acceleration of activation and quality of carbon?
2) Table 6 contains a lot of adsorption models, but it is unclear, what models are appropriate for activated carbons of interest. Does suitable adsorption equation depend on chemical reagent and heating method?
Author Response
The review is dedicated to researches of relations between pyrolysis conditions (heating, reagents) and characteristics of activated carbon.
The paper is organized and written quite well. Nevertheless, I have some questions.
1) It can be concluded from fig. 5 that microwave heating leads to worse quality of activated carbon: surface area and adsorption capacity are less than at convective heating. So question arises: does microwave heating allow to enhance activation process? What is weighted profit of microwave systems in terms of acceleration of activation and quality of carbon?
RESPONSE
These are interesting technical questions. Microwave process, in some of the studies referred to in this review were not good as compared to conventional method. However, this could be due to the choice of reactor, radiation power and other production challenges. The benefits of microwave process are evident, namely; speedy rate of activation, energy saving and improved carbon yield.
2) Table 6 contains a lot of adsorption models, but it is unclear, what models are appropriate for activated carbons of interest. Does suitable adsorption equation depend on chemical reagent and heating method?
RESPONSE
The adsorption equations are independent of the chemical reagents and heating methods. The models depend on the thermodynamic state defined by pressure and temperature during adsorption. Some adsorption analysis must be carried out to determine the appropriate model, there is therefore no best model for any process. As a result, in this review the descriptions are itemised to provide information on the relevance of each model and indicate how they can fit an adsorption process.
Reviewer 2 Report
Nowadays, the idea of converting agro-industrial waste into energy and other useful products represents a research area with great potential and opportunities, but the technologies used for their exploitation must be environmentally friendly and sustainable from the economically point of view
This review attempts to evaluate multiple activating agent effects and make generic comparison and assessment of multiple agricultural wastes, while outlining current process challenges. In particular, this study, considers the effectiveness of some of the chemical activating agents used for AC production under various production conditions and also looks at the effects on the product characteristics, of surface functional groups and overall adsorption efficiency for several applications. Comparative analyses of production methods are also examined to suggest more appropriate pathways.
Overall the manuscript addresses an interesting topic and prior to publication some minor improvements are required.
The article is generally well written, the English language is appropriate and understandable, only minor spell check being required. Also, the sound is not very scientific and a correction by an experimented speaker would be recommended. Please provide a better-quality version for Figure 1, 4 and 5
Author Response
Nowadays, the idea of converting agro-industrial waste into energy and other useful products represents a research area with great potential and opportunities, but the technologies used for their exploitation must be environmentally friendly and sustainable from the economically point of view
This review attempts to evaluate multiple activating agent effects and make generic comparison and assessment of multiple agricultural wastes, while outlining current process challenges. In particular, this study, considers the effectiveness of some of the chemical activating agents used for AC production under various production conditions and also looks at the effects on the product characteristics, of surface functional groups and overall adsorption efficiency for several applications. Comparative analyses of production methods are also examined to suggest more appropriate pathways.
Overall the manuscript addresses an interesting topic and prior to publication some minor improvements are required.
The article is generally well written, the English language is appropriate and understandable, only minor spell check being required. Also, the sound is not very scientific and a correction by an experimented speaker would be recommended. Please provide a better-quality version for Figure 1, 4 and 5
RESPONSE
The grammatical errors have been corrected and appropriate scientific terms included.
Figure 1, 4 and 5 have been redrawn.
Reviewer 3 Report
The manuscript represents a comprehensive review on the topic of generating activated carbon from waste biomass. After a good introduction into the importance of the subject the authors cover a a number of relevant subtopics that are discussed in sufficient detail to provide the interested reader with a solid foundation of relevant developments and the reference section will aid the reader with further details. Suitable graphics and tables are included to summarise and visualise larger sets of data or structural information.
This reviewer would recommend including a table of contents section at the beginning. It would also be advisable to improve the quality of drawings such as Figure 1 and Figure 4 with respect to using a uniform and more legible drawing style. This reviewer noticed that the language and grammar leading up to subsection 6 is oftentimes substandard making the reading difficult. It is suggested to have the entire manuscript checked by a qualified native speaker or language editing service. Some minor mistakes are: line 128 "fewer number" = lower number, line 133 and elsewhere, separate numbers and their units by a space; line 117 "aldehydic and carbonyl groups", aldehydes are carbonyls too, aldehydic is not used as an adjective; line 223 rephrase "but in for large production of large quantities"; line 337 " the sensitivity of the cellulose linkage towards...", line 471 "metalic ions" = metal ions; etc.
If the authors can comment on these and make corrections accordingly, this reviewer would be able to recommend acceptance of this manuscript as review in Sustainability.
Author Response
The manuscript represents a comprehensive review on the topic of generating activated carbon from waste biomass. After a good introduction into the importance of the subject the authors cover a a number of relevant subtopics that are discussed in sufficient detail to provide the interested reader with a solid foundation of relevant developments and the reference section will aid the reader with further details. Suitable graphics and tables are included to summarise and visualise larger sets of data or structural information.
This reviewer would recommend including a table of contents section at the beginning. It would also be advisable to improve the quality of drawings such as Figure 1 and Figure 4 with respect to using a uniform and more legible drawing style. This reviewer noticed that the language and grammar leading up to subsection 6 is oftentimes substandard making the reading difficult. It is suggested to have the entire manuscript checked by a qualified native speaker or language editing service.
RESPONSE
Figure 1, 4 and 5 have been redrawn. The grammar is modified and carefully checked. Other errors identified in line 117, 133, 223, 337 and 471 have been corrected.
The table of content is an excellent idea; however, the template does not support it, perhaps the editor may approve a table of content.